# Reliable Off-Resonance Correction in High-Field Cardiac MRI Using Autonomous Cardiac B_0_ Segmentation with Dual-Modality Deep Neural Networks

**DOI:** 10.3390/bioengineering11030210

**Published:** 2024-02-23

**Authors:** Xinqi Li, Yuheng Huang, Archana Malagi, Chia-Chi Yang, Ghazal Yoosefian, Li-Ting Huang, Eric Tang, Chang Gao, Fei Han, Xiaoming Bi, Min-Chi Ku, Hsin-Jung Yang, Hui Han

**Affiliations:** 1Biomedical Imaging Research Institute, Cedars-Sinai Medical Center, Los Angeles, CA 90048, USA; xinqi.li@mdc-berlin.de (X.L.); archana.malagi@cshs.org (A.M.); chia-chi.yang@cshs.org (C.-C.Y.); li-ting.huang@cshs.org (L.-T.H.); erictang0220@gmail.com (E.T.); 2Berlin Ultrahigh Field Facility (B.U.F.F.), Max Delbrück Center for Molecular Medicine in the Helmholtz Association (MDC), 13125 Berlin, Germany; min-chi.ku@mdc-berlin.de; 3Krannert Cardiovascular Research Center, Indiana University School of Medicine, Indianapolis, IN 46202, USA; yuhenghuang@g.ucla.edu (Y.H.); gyoosefi@iu.edu (G.Y.); 4Bioengineering, University of California, Los Angeles, Los Angeles, CA 90095, USA; 5MR R&D Collaborations, Siemens Medical Solutions Inc., Los Angeles, CA 90048, USA; chang.gao@siemens-healthineers.com (C.G.); fei.han@siemens-healthineers.com (F.H.); xiaoming.bi@siemens-healthineers.com (X.B.); 6Department of Radiology, Weill Medical College of Cornell University, New York, NY 10065, USA

**Keywords:** cardiac MRI, B_0_ shim, B_0_ field map, dual modality image segmentation

## Abstract

$B_{0}$ field inhomogeneity is a long-lasting issue for Cardiac MRI (CMR) in high-field (3T and above) scanners. The inhomogeneous $B_{0}$ fields can lead to corrupted image quality, prolonged scan time, and false diagnosis. $B_{0}$ shimming is the most straightforward way to improve the $B_{0}$ homogeneity. However, today’s standard cardiac shimming protocol requires manual selection of a shim volume, which often falsely includes regions with large $B_{0}$ deviation (e.g., liver, fat, and chest wall). The flawed shim field compromises the reliability of high-field CMR protocols, which significantly reduces the scan efficiency and hinders its wider clinical adoption. This study aims to develop a dual-channel deep learning model that can reliably contour the cardiac region for $B_{0}$ shim without human interaction and under variable imaging protocols. By utilizing both the magnitude and phase information, the model achieved a high segmentation accuracy in the $B_{0}$ field maps compared to the conventional single-channel methods (Dice score: 2D-mag = 0.866, 3D-mag = 0.907, and 3D-mag-phase = 0.938, all *p* < 0.05). Furthermore, it shows better generalizability against the common variations in MRI imaging parameters and enables significantly improved $B_{0}$ shim compared to the standard method (SD($B_{0}$Shim): Proposed = 15 ± 11% vs. Standard = 6 ± 12%, *p* < 0.05). The proposed autonomous model can boost the reliability of cardiac shimming at 3T and serve as the foundation for more reliable and efficient high-field CMR imaging in clinical routines.

## 1. Introduction

Cardiac Magnetic Resonance Imaging (CMR) represents a pivotal advancement in cardiac care, offering a comprehensive and non-invasive approach to assessing heart structure, function, and myocardial tissue characterization. CMR utilizes the magnetic resonance signal from water protons in the heart to provide pathologically sensitive signals without exposing patients to ionizing radiation. This allows the application of CMR to extend beyond mere anatomical visualization and play a crucial role in the evaluation of myocardial perfusion, ventricular contractility, myocardial viability, and myocardial tissue composition [1,2,3,4,5], making it the preferred modality in the diagnosis and management of a variety of cardiac conditions, such as cardiomyopathies, heart failures, and congenital heart diseases [6,7,8,9,10].

### 1.1. $B_{0}$ Inhomogeneity Effect on 3.0T CMR

Since the FDA approved the use of 3.0T scanners for whole-body clinical applications in 2002, the adoption of high-field scanners has been rising rapidly, particularly for neuroimaging. In general, 3.0T provides higher SNR, spatial resolution, and reduced scan time than 1.5T. In some facilities, 3.0T may be the only available field strength. However, 3.0T adoption for CMR has been relatively slow. For CMR, the superior SNR at 3.0T provides high potential to facilitate accelerated imaging with enhanced spatial and temporal resolution, which can be further optimized through techniques such as compressed sensing and deep learning [11,12,13]. The increased T1 relaxation times at 3.0T augment T1-weighted imaging, improving the diagnostic quality of late gadolinium enhancement (LGE) and first-pass perfusion methods [14]. This leads to improved myocardial tissue characterization. Moreover, the amplified T2* contrast inherent to 3.0T MRI allows for a more effective assessment of iron deposition [11,12], hemorrhage [13], and oxygen consumption [14,15], which provides critical information for comprehensive evaluation of cardiac pathology. Furthermore, the improved spectral separation at 3.0T enhances metabolic imaging, magnetization transfer, and magnetic resonance spectroscopy imaging, allowing for refined chemical exchange saturation transfer (CEST) imaging [15,16] and more effective fat suppression in coronary imaging. Collectively, these advantages have the potential for 3.0T systems to elevate the diagnostic capabilities of CMR.

However, despite its numerous benefits, high-field cardiac MRI poses unique challenges that hinder its wider clinical adoption. A major challenge for 3.0T CMR remains the increased $B_{0}$ inhomogeneity from the amplified main field, particularly at the tissue–air interface due to susceptibility variations [17,18,19].

$B_{0}$ field homogeneity is critical for optimal CMR imaging, particularly when leveraging the high signal-to-noise (SNR) benefits of 3.0T systems. Steady-state free precession (SSFP), a cornerstone CMR sequence at 1.5T due to its rapid acquisition and high SNR, encounters significant challenges at 3.0T. $B_{0}$ inhomogeneity at this higher field strength results in substantial signal variability and banding artifacts [17,18], as shown in Figure 1. Similarly, echo-planar imaging (EPI)—despite its efficiency—is vulnerable to $B_{0}$ inhomogeneity, leading to image distortion and signal loss at 3.0T [20]. This inhomogeneity further undermines the consistency of T2*-based sequences [2] and the effectiveness of fat suppression techniques, both imperative for detailed myocardial and coronary artery visualization [21]. To capitalize on the SNR advantages of 3.0T CMR, reliable $B_{0}$ shimming techniques and sequence adaptations are necessary to acquire high-quality diagnostic images.

### 1.2. Cardiac $B_{0}$ Shimming to Improve Image Quality of High-Field CMR

Active $B_{0}$ shimming is the most direct way to correct for $B_{0}$ field inhomogeneities [22,23,24]. By creating a correction $B_{0}$ field from shim coils, $B_{0}$ shimming adjusts the static magnetic field across the imaging volume to ensure field uniformity. Recent advancements in shimming technology have led to the development of advanced shim hardware, like RF coils with integrated $B_{0}$ shimming [24,25], capable of generating high-order shim fields to correct for the unique off-resonance patterns in the heart. However, the heart’s shape, location, and tissue composition make accurately measuring the $B_{0}$ field challenging. In today’s standard clinical practice, manual selection of a shim box is used to identify shim volumes for $B_{0}$ off-resonance estimation [26]. Yet, there are common confounders, such as the false inclusion of non-cardiac off-resonance sources (e.g., liver and chest wall) and fat-induced chemical shifts in the shim box. These confounders can significantly degrade the accuracy of shim field estimation, leading to failed $B_{0}$ shimming and further degradation of the image quality [19,27,28], as shown in Figure 2. To facilitate accurate $B_{0}$ shimming, it’s critical to precisely delineate the heart region and ensure the shim coils can generate the optimized cancellation shim field in the heart and improve cardiac image quality [27]. While prior works have shown manual contouring of the cardiac region can improve the shim robustness and shimming accuracy, manually contouring the region of interest in CMR images is time-consuming and expensive, making it impractical for clinical settings. An autonomous and reliable segmentation method for CMR $B_{0}$ maps is desired to improve $B_{0}$ shimming robustness and facilitate reliable high-field CMR scans.

### 1.3. State-of-the-Art CMR Segmentation Models Are Not Optimized for $B_{0}$ Field Maps

Image segmentation has been a crucial task in CMR applications. Manual segmentation is time-consuming and labor-expensive so conventionally there are some semi-automatic techniques such as threshold [29], region grow [30] and contour-based methods [31].

With the rapid evolution of deep learning techniques, multiple automatic segmentation models have been proposed in recent years [32,33,34,35,36,37]. The U-Net structure, characterized by a symmetric encoder and decoder with skip connections, has succeeded greatly across various medical imaging domains [35]. In the encoder, multiple convolutional and down-sampling layers are used continuously to extract the feature information. The decoder up-samples the extracted feature within a large receptive field to the input resolution to enable pixel-level semantic prediction. The skip-connection between each layer of encoder and decoder helps mitigate the information degradation during down-sampling and up-sampling [38]. Following this elegant design, model structures such as 3D U-Net [39], U-Net++ [36], nnU-Net [33] have been developed. The nnU-Net framework, in particular, is an automated configuring framework that provides a standardized and efficient pipeline with highly accurate segmentation results [33]. In the evolving landscape of deep learning-based automatic segmentation methods, its application in CMR images has also emerged as a pivotal tool for CMR image analysis [40,41]. Although convolutional neural networks have achieved huge success in segmentation tasks, transformer based methods and generative models have been popular recently due to their great success in other computer vision tasks. For example, BerDiff [42] utilized the conditional Bernoulli Diffusion model, ViT-FRD [43] that combines a visual transformer and a CNN through knowledge refinement, and Swin-UNETR [32] that incorporated the vision transformer into a U-structure.

Segmentation models have traditionally been designed for single-channel signal magnitude variations. However, regarding $B_{0}$ field maps in MRI, the single-channel approach faces limitations due to their unique contrast characteristics. Standard MRI segmentation models do not work well with $B_{0}$ field maps as these images are often proton-density weighted and have low soft tissue contrasts. The challenge is particularly evident in cardiac $B_{0}$ field maps, where the heart and liver are in proximity and have unclear boundaries. As a result, these models struggle to provide accurate cardiac boundary delineation in cardiac $B_{0}$ field maps for shim field derivation. Since MRI is based on the magnetic resonance of water protons, MRI images consist of both magnitude and phase signal data. Phase images in MRI provide crucial spatial information by reflecting the chemical composition and local field inhomogeneity within the tissue. The phase images of the heart and liver can exhibit distinctive features due to local frequency changes influenced by their respective location, shape, and orientation in relation to both the air-filled space and the $B_{0}$ direction. However, this aspect of MRI is often overlooked despite its potential for improving the reliability and robustness of segmentation algorithms.

In this study, we hypothesize that by incorporating phase images, which are inherently sensitive to organ boundaries, segmentation accuracy can be significantly enhanced. The main contributions are:We developed a dual-channel CNN model to improve cardiac segmentation for $B_{0}$ shimming in high-field CMR by combining magnitude and phase images.We thoroughly evaluated the performance of the proposed model under different imaging parameters and compared it with state-of-the-art medical image segmentation techniques. Besides, we demonstrated the generalizability of dual-channel module on different existing models to improve the performance.We further demonstrated the application of this dual-channel segmentation model in providing the foundation of high-quality $B_{0}$ shimming in the heart.

## 2. Methodology

### 2.1. Image Preprocessing

Given that various imaging parameter sets were employed in experiments, a standardized image pre-processing routine was applied to calibrate the data. The data pre-processing included the following steps:(1)*Background removal*: The raw data contained some redundant air introduced during the image acquisition and reconstruction. As a first step, Otsu’s method [44] was derived from the magnitude maps with number of threshold values equal to 2. The rough mask was generated based on the threshold level and followed by a post-processed operation using morphological closing. The structuring element was a disk-shape one defined by the resolution of the image.and applied to both the magnitude and phase map. It helped effectively segregate the region of interest from extraneous air.(2)*Resolution and FOV Alignment*: The voxel spacing within our acquired data was heterogeneous. The large spacing might cause the loss of detailed information, while the small spacing requires a larger computational budget. To reconcile this, we established a target voxel spacing based on the median spacing observed across all subjects for each axis. Given the anisotropic nature of our dataset, we resampled all images to the uniform target voxel spacing using third-order spline interpolation. Subsequently, images were either cropped or padded to match the dimension at the center region, if necessary.(3)*Noise Standardization*: All images were normalized based on mean and standard deviation values per case. The normalization step ensured all data conformed to a consistent scale and distribution.(4)*Dataset split*: The T1-w data set, including 54 subjects, was randomly partitioned into a training set (comprising 40 volumes) and a test set (comprising 14 volumes). Additionally, 10 PD-w volumes from subjects not included in the training set were reserved for an independent test set to validate the model’s generalizability across varied imaging protocols.

### 2.2. Model Architecture

In this study, we proposed a two-channel segmentation model built based on the nnU-Net, integrating both the magnitude and phase information for heart segmentation in CMR images. As shown in Figure 3, the magnitude map cannot provide a clear contour for the heart region, while the phase map can provide additional information to delineate the heart. The general pipeline is shown in Figure 4. The magnitude and phase map will be pre-processed and concatenated into a 4D matrix with the additional channel dimension. Then we trained three different models, namely 2D-mag-net, 3D-mag-net, 3D-mag-phase-net, where 2D-mag-net was a 2D U-Net model, 3D-mag-net was a single channel nnU-Net model only using magnitude information and 3D-mag-phase-net was a dual-channel nnU-Net based model using magnitude and phase information. As we used the cross-fold validation in training, during the inference, we ensembled the softmax probabilities of 5 folds to predict the segmentation. Besides, we applied the connected component-based post-processing [45] to eliminate the obvious false positives and generate the final prediction.

The proposed dual-channel nnU-Net-based model followed the original U-Net [35] structure with 5 encoder and decoder layers. Each layer included two convolutional blocks. Within each convolutional block in the encoder and decoder, we incorporated a 3 × 3 × 3 convolution with stride 2, an instance normalization and a leaky ReLU nonlinearity with a negative slope of 0.01. Notably, we utilized the leaky ReLU activation function, which differs from standard ReLU because of its smaller slope for negative values. In the encoder, strided convolutions with a stride size of 2 were employed for down-sampling, while the transposed convolution was used in the decoder to up-sample the feature map. We employed the dice loss function, defined as follows: (1)$$L_{Dice}=\;-\;\frac{2\sum_{i}o_{i}y_{i}}{\sum_{i}o_{i}+\sum_{i}y_{i}},$$
where $o_{i}$ represents the voxel’s value from the labeled volume and $y_{i}$ represents the voxel’s value from the predicted volume.

### 2.3. Training Strategy

Previous studies demonstrated that a large patch size is important for model training, as a small batch size leads to noisier gradients during the training, and a larger patch size allows the aggregation of more information [46]. To accommodate large patch sizes, we maintained a modest batch size of 2, with the patch size tailored for different configurations. The stochastic gradient descent with Nesterov momentum and an initial learning rate of 0.01 was used to learn weights. Each network was trained for 1000 epochs, with each epoch consisting of 250 mini-batches. To prevent the drastic reduction of the number of samples that can be used for learning, we implemented cross-validation during training, using different portions of the training set as train and validation data, allowing better evaluation of the performance. The training set was divided into *k* (*k* = 5) smaller sets and one subset as validation set was used each time during training. During the inference, *k* models were averagely ensembled to predict the segmentation.

### 2.4. Data Augmentation

In medical images, previous studies have proved that data augmentation is critical given the limited data samples and the complexity of medical images. Several data augmentation strategies were applied on the fly by probability during the training phase: rotation, scaling, Gaussian noise, simulation of low resolution, and mirroring. We found that scaling was significant in our case due to the nature of different patient sizes. Comprehensive details are provided in the Experiments section. The augmentation was implemented using TorchIO [47]. In addition to the standard augmentation parameters, the apparent size of the subjects is a common variation in medical images (Figure 5). Although the problem of object size variation has been extensively studied in machine learning algorithms, especially for object detection tasks [48], its application in medical imaging models is more complicated with the limited field of view and partially covered body parts. To mitigate this issue, we further simulated the diverse patient sizes. This was accomplished by either cropping or padding the image from its center and rescaling it to induce the FOV coverage effect.

### 2.5. Evaluation Metrics

We measured the performance based on two key metrics: (1) Dice Score. The Dice score computed the overlap between two volumes, varying from 0 (mismatch) to 1 (perfect match). It is defined as: (2)$$D\left(A,\;B\right)=2\frac{A\;\cap B}{A+B},$$
where *A* and *B* correspondingly denote the sets of heart voxels in ground-truth and predicted volumes. (2) 95% Hausdorff Distance. The Hausdorff distance (HD) calculated the maximum distance between two volumes. It is defined as: (3)$$H\left(A,\;B\right)=\max_{A,B}\left\{d_{AB},d_{BA}\right\}=\max_{A,\;B}\left\{\max_{x\in A}\min_{y\in B}d\left(x,\;y\right),\;\max_{y\in B}\min_{x\in A}d(x,\;y)\right\},$$
where *d* represents Euclidean distance. To calculate the 95% HD, the calculation is based on the 95th percentile of the distances between boundary points in *A* and *B*. Lower values of the 95% HD indicate superior segmentation performance. (3) Jaccard Index. The Jaccard Index or named as Jaccard similarity coefficient was defined as the Intersection over Union (IoU) between the ground-truth and predicted results [49].

### 2.6. Statistical Analysis

All the data are represented as mean ± standard deviation (SD). We performed paired *t*-test for paired comparisons and repeated measures of Analysis of Variance (ANOVA) for 3 way comparisons (2D-mag, 3D-mag, 3D-mag-phase), using the Scipy [50] and Statsmodels [51] in Python 3.10. The post-hoc analysis based on the pairwise *t*-tests with Bonferroni correlation.

## 3. Results

Three neural networks (2D-mag-net, 3D-mag-net, 3D-mag-phase-net) were implemented using PyTorch based on nnU-Net. These networks were initially trained on a dataset including 40 T1-w CMR images. Subsequently, models were applied to test data without fine-tuning or further training. The developed model is available at https://github.com/lixinqi98/DynamicShim, accessed on 18 January 2024.

### 3.1. Datasets

All images were acquired in a 3T MR systems (Biograph mMR, Siemens Medical Solutions, Erlangen, Germany). Healthy volunteers (*n* = 64) were recruited per the protocol reviewed and approved by the institutional review board (IRB), No.23469. Every participant was competent, provided written informed consent, and had no history of coronary artery disease, lung disease, abnormal cardiac rhythm and rate, kidney or liver disease, and was not contraindicated for cardiac MR examinations (i.e., they completed a detailed cardiac MRI questionnaire). To explore the generalizability of the proposed method, $B_{0}$ field maps were acquired with two sets of imaging parameters. One set (T1 weighted $B_{0}$ maps, T1-w) included 54 different healthy subjects’ cardiac volume with TE1/TE2 = 1.31/3.53 ms; Flip angle = 16°; FOV = 400 × 300 × 250 ${\mathrm{mm}}^{3}$; Spatial resolution = 3.57 × 3.5 7× 5.2 ${\mathrm{mm}}^{3}$. Echo spacing = 2.1 ms. Another set (Proton density weighted $B_{0}$ maps, PD-w) includes 10 healthy subjects acquired with TE1/TE spacing = 1.42/2.01 ms; number of echoes = 6; flip angle = 8°; FOV = 300 × 300 × 120 ${\mathrm{mm}}^{3}$; spatial resolution = 1.56 × 1.56 × 5 ${\mathrm{mm}}^{3}$. The ground-truth segmentations were manually drawn by experts(MRI scientist and Radiologist). All the data follows the image preprocessing in Section 2.1, matched to median resolution 3.57 × 3.57 × 5.2 ${\mathrm{mm}}^{3}$ and size of 96 × 96 × 40. According to the data augmentation in Section 2.4, the summary of data before and after augmentation can be found in Table 1.

### 3.2. Model Performance

We tested the models’ performance first on images with identical imaging parameters (T1-w images) to the training data set for assessing the segmentation ability against anatomical variations between subjects. For the dice score, the 2D-mag-net, 3D-mag-net, and 3D-mag-phase-net was 0.87 ± 0.04, 0.91 ± 0.02, and 0.94 ± 0.04, respectively. For 95% HD, 2Dmag-net, 3D-mag-net, and 3D-mag-phase-net was 11.20 ± 5.90, 7.78 ± 4.62, and 6.20 ± 3.61 mm, respectively. For Jaccard index, 2Dmag-net, 3D-mag-net, and 3D-mag-phase-net was 0.76 ± 0.06, 0.83 ± 0.04, and 0.87 ± 0.07, respectively. As illustrated in Figure 6, under the null hypothesis that the predictions made by these models share the same distribution, a significant enhancement in the dice score is presented in the dual channel model. This indicated that our model (3D-mag-phase-net) outperformed other models with the help of additional phase information.

Furthermore, as we applied the cross-validation during the training, we evaluated the consistency across different folds in our cross-validation approach. We generated the segmentation employing the model from each fold. As illustrated in Figure 7, the performance of various models across each fold was examined. The bar plot demonstrated that, in most of the folds, our 3D-mag-phase model consistently exhibited superior performance. This finding underscores the robustness and reliability of the proposed method.

In Figure 8, we presented the training progress of different models. Notably, the 2D-mag-net converged fastest due to a larger batch size. The larger batch size also leads to a more stable training process. However, the performance of the 2D-mag-net, as previously demonstrated, suggested a potential overfitting of the training data. The batch size of the two 3D models is the same. Our proposed model converged faster than the 3D-mag-net and maintained a more stable training process.

### 3.3. Generalizability Analysis

To assess the generalizability of our models, we conducted the following ablation studies.

#### 3.3.1. SNR Variations

We investigated the robustness of different models to the changes in the Signal-to-Noise Ratio (SNR). We simulated the potential noise during the data acquisition by adding Gaussian noise. The noise was sampled from a normal distribution with a mean of 0 and standard deviations ranging from 0.01 to 0.05. As shown in Figure 9, the 3D models demonstrated a significantly higher degree of robustness in the presence of noise, whereas the performance of the 2D-mag-net exhibited a significant decline. Notably, our 3D-mag-phase-net consistently outperformed other models, and the standard deviation of the performance was consistently smaller than the 3D-mag-net throughout all SNR levels.

#### 3.3.2. Imaging Protocol Variations

In addition to the training dataset, we conducted experiments on field maps acquired with a different imaging protocol to test the model’s performance against the variation of image contrast and image resolution. Proton density-weighted (PD-w) high-resolution images were acquired from 10 healthy subjects. Furthermore, since field maps can be acquired under different breath-holding states in different clinical practices, we further collected images under end-inspiration and end-expiration to investigate the influences of the respiratory position on the model performance. We predicted the contours using the trained models without further fine-tuning or training on the new dataset. The models’ performance against the imaging protocol variation is compared in Figure 10. The proposed 3D-mag-phase-net showed significantly improved dice scores compared to the magnitude-only models under the new dataset with different image contrast and resolution. The summary results was shown in Table 2. The 95% HD shows a similar trend. (In end-expiratory, 2D-mag-net’s average 95%HD and SD was 6.610 ± 0.921, 3D-mag-net was 6.019 ± 0.792, and 3D-mag-Phase-net was 5.775 ± 1.486. In end-inspiratory, 2D-mag-net’s average 95%HD and SD was 7.824 ± 2.421, 3D-mag-net was 6.396 ± 1.080 and 3D-mag-phase-net was 5.922 ± 1.837.) Furthermore, consistent results are shown in the end-expiratory or end-inspiratory cycle, demonstrating the models’ robustness against respiratory motion.

#### 3.3.3. Comparisions between Model Architectures

To evaluate the ability of different model structures to utilize the phase information, we extended our work to incorporate the dual-channel into other existing deep learning-based segmentation models. Specifically, we implemented the naïve UNet and the Swin UNETR models based on the MONAI [52] framework. The training strategy remained consistent with our prior description, and we evaluated the segmentation results on all the same testing datasets. In addition, we finetuned a state-of-the-art transformer-based single-channel model (SAM-Med3D [53]) based on our training dataset to test its performance on the task of field map segmentation. The models’ performance is shown in Table 3. In all U-net-based models (U-Net, Swin UNETR, and ours (3D-mag-phase-net)), the magnitude-phase model showed improved dice scores compared to the magnitude-only models. This indicates that the additional information from phase maps can be extracted regardless of the model structures. In addition, the proposed 3D-mag-phase-net outperformed all other models (all *p* < 0.05) and showed an accurate segmentation result across the different imaging protocols. It is worth noting that, in recent studies, transformer-based models have been recognized to outperform convolution-based models when trained with large heterogeneous datasets [54]. However, due to the limited training data size, the SAM-Med3D model and Swin UNETR did not show the desired performance on this task in our study. In contrast, despite the limited training data, our model reliably performs and shows significant advantages in model convergence.

#### 3.3.4. Cardiac $B_{0}$ Shimming Experiments and Performance Comparison

The $B_{0}$ shimming ability of the proposed method is evaluated and compared to the standard manual-selected box shim volume in Figure 11. $B_{0}$ shimming was derived using the 2nd-order spherical harmonic shim coils that are equipped with state-of-the-art 3T clinical scanners. Representative images of the cardiac region from axial and coronal views demonstrated the shimming performance was improved with the proposed autonomous shimming pipeline and is shown in Figure 11A. To validate the visualized improvement, the shimming performance was evaluated quantitatively using standard deviation (SD) and interquartile range (IQR) of the $B_{0}$ field in the heart and compared to the standard manual box shim in Figure 11B. Significantly more homogeneous $B_{0}$ from the proposed model is presented in SD and IQR and reflect a more reliable performance of $B_{0}$ shimming without human interaction. Cardiac $B_{0}$ SD (SD($B_{0}$Shim)/SD($B_{0}$)) decreased 15 ± 11% using our proposed autonomous shimming, while 6 ± 12% using standard manual shimming. For IQR, IQR($B_{0}$Shim)/IQR($B_{0}$) decreased 21 ± 12% using our autonomous ROI, while 14 ± 18% using standard manual box volume (all *p*-value < 0.05).

#### 3.3.5. Ablation Study

To better understand the contribution of each part, except comparing the 2D-mag, 3D-mag and 3D-mag-phase net, we further investigated the following components for ablation: instance normalization and augmentation. For each setting we re-trained the model and evaluate the performance using cross-fold validation. The performance on validation set was quantified on each component. In our implementation, we utilized the instance normalization instead of batch normalization after each convolution operation. In our experiments, the change from batch normalization to instance normalization will decrease the average validation dice score from 0.93 to 0.91 but drastically accelerate the process. The small batch size (batch size of 2 in our implementation) limited the batch normalization’s ability to speed up and stablize training. While the instance normalization can deal with the noiser mean and variance when the batch size is very small. In our proposed model, we finally chose the instance normalization, as the segmentation accuracy was more important in shimming application. Additionally, we experimented the importance of data augmentation, we found that without scale augmentation, the dice score decreased averagely 2.15% on the validation set, which is consistent with our observation in Section 2.4 that various patient size in test set will affect the model performance.

## 4. Discussion

In this paper, we explored the integration of magnitude and phase information to enhance the accuracy of 3D segmentation models for CMR field maps and its ability to improve $B_{0}$ shimming compared to the standard manual shimming pipeline. We evaluated the model and demonstrated the robustness and generalizability of the proposed dual-channel model using CMR field maps acquired with different contrast weighting and imaging parameters. The proposed 3D-mag-phase-net, built based on the nnU-Net structure, successfully harnessed the complementary information from phase maps, especially benefiting the segmentation accuracy in regions with impaired tissue contrast in the magnitude images. It demonstrates reliable performance in real-world data through data variations commonly presented in the clinical setting.

Previous research has indicated that a conventional U-net model has the potential to predict cardiac contour and aid in cardiac $B_{0}$ shimming [55]. However, this approach has only been tested on 1.5T scanners, and its efficacy in higher field scanners remains untested. Moreover, the previous study was performed using fixed imaging protocols and did not explore the possibility of performance variation between field map acquisition parameters, which means that the model’s generalizability is unclear.

In this study, we developed a dual-channel model combined with an advanced nnU-Net architecture to accommodate the potential variation of field map acquisition. We tested the model at 3.0T, where the $B_{0}$ off-resonance is stronger and affects its daily application in the clinical setting. The integration of magnitude and phase images allows for structural and functional insights, making it a powerful tool for the segmentation task in MR images. Notably, in segmenting the cardiac field map, a major challenge for conventional methods is to separate the connected heart and liver at the apical portion of the left ventricle. This is particularly important for $B_{0}$ shimming as strong off-resonance artifacts are commonly presented in this region due to the unfavorable heart-lung anatomy. Because the conventional magnitude images (both T1 or proton-density weighted) exhibit similar signal intensity between the heart and the liver. The magnitude-only segmentation methods often fail in this region and significantly affect the shim results. In contrast, phase maps showed strong phase differences between the organs, reflecting their local frequency changes influenced by their respective location, shape, and orientation in relation to both the air-filled space and the $B_{0}$ direction, as shown in Figure 12. The phase map provides clear delineation at the heart-liver interface, which can facilitate reliable segmentation for the heart.

In addition to the organ boundaries, tissue composition is critical for field map segmentation. Particularly, the epicardial fat can cause contrast variations in the T1w image and introduce contrast changes between imaging parameters, in Figure 12. This can compromise the generalizability of the magnitude-only models. The phase maps provide a consistent frequency profile of the fat signal, which can assist in identifying the fat tissue and keep the consistency of cardiac field map segmentation.

In our study, the generalizability of the models between imaging parameters is tested under different MRI acquisition protocols. We showed the ability of the proposed model to maintain accurate segmentation capability under different SNR, resolution, and MRI imaging contrasts. In the conventional single-channel techniques, the change in imaging contrast is usually a domain-transferring task in segmentation models. Fortunately, the phase maps of MR field maps with multi-echoes are quantitative and resilient to imaging parameter changes. This provides a consistent domain for segmenting the target organs in MR field maps.

One potential trade-off of this integration is the dependence on the quality of the phase maps. Artifacts caused by phase wrapping and motion-induced phase errors can degrade the performance of the segmentation. These challenges in the phase domain will propagate the errors and even mislead the segmentation results. Mitigating such artifacts is crucial, and future work may also include developing more advanced algorithms for phase unwrapping and motion correction to fully leverage the integrated information [28,56,57]. Another consideration is the additional computational demand of the dual-channel model. The processing of additional channels inherently requires more computation resources, such as larger RAM to feed the data, potentially limiting its clinical applicability. The current model is compatible with a single workstation equipped with an NVIDIA GeForce RTX 4090 GPU with 24 GB RAM. Using a relatively small batch size and median patch size as mentioned in the methods, we can successfully deploy the model in a state-of-the-art scanner’s host computer. To enable broader applications, more investigation into computational efficiency without drastically compromising performance can be done in the future. This can help the development of more streamlined clinical applications in scanners with less computation powers.

The U-net structure based on convolutional blocks has shown its success in various medical image-related tasks [58], with the help of its U-structure, to capture the semantic information within the image. That being said, the transformed-based backbone utilizing the attention mechanism is catching popularity in the computer vision field [59]. The attention mechanism allows the transformer model capture the long-range and global context information while convolutional layers usually local semantic information [60]. Although transformer-based models showed promising results [32,61,62] in medical image analysis, due to the large model size, the networks are more difficult to train and require larger training data [63]. A common practice for the transformed-based network is to start from the pre-trained model on large-scale datasets [54], which allows for fine-tuning specific tasks with smaller data input. However, fine-tuning a large model is not trivial work [64]; it requires careful hyperparameter tuning, including learning rates, batch sizes, and normalization techniques, making the finetuning of the transformer models on the limited medical imaging datasets challenging. In this study, we found that our fine-tuning of the SAM-Med3D model [53] performed much worse than the proposed U-net-based model in the test dataset. This might be due to the unique imaging contrast in the $B_{0}$ field maps and the limited size of the training data, which does not provide enough diversity and complexity for the model.

The application of our proposed model in cardiac $B_{0}$ shimming is shown in a high-field in-vivo experiment by creating autonomous contours of the field maps. Our data demonstrated that the model’s ability to produce accurate, motion-robust contours on cardiac field maps quickly could significantly enhance the $B_{0}$ homogeneity with the standard clinical shimming hardware. Furthermore, it is worth noting that the shim volume accuracy is particularly important for the fast-developing multi-coil shimming hardware and combined shim-RF coils [24,25,65,66,67]. Because of the increased flexibility of the shimming fields, an erroneous shimming ROI can lead to strong overfitting to the false $B_{0}$ field and corrupt the overall robustness of the procedure. The developed autonomous shimming pipeline can be a crucial step for implementing high-order shimming hardware and its clinical applications.

## 5. Conclusions

Accurate $B_{0}$ shimming is crucial for successful high-field cardiac MRI studies. The developed dual-channal model incorporating phase and magnitude information has achieved high segmentation accuracy in cardiac $B_{0}$ field maps that are insensitive to imaging acquisition protocol changes. The integration of the developed autonomous pipeline into clinical scanners could serve as the foundation for reliable high-field CMR imaging and widern its clinical adoption. Furthermore, future works to combine the developed method with high-order shimming hardware can further boost the $B_{0}$ homogeneity and enable advanced imaging contrast for a wide range of cardiovascular diseases.

## Figures and Tables

**Figure 1 bioengineering-11-00210-f001:**
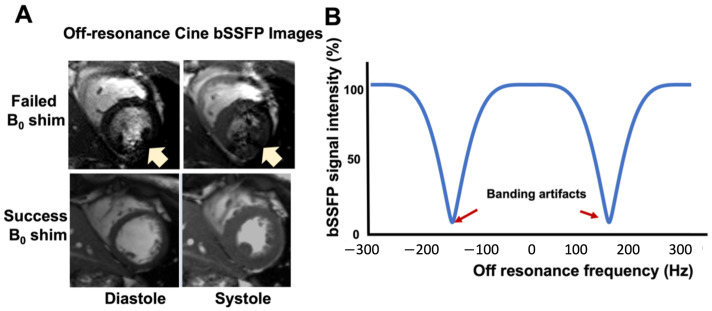
Banding artifacts in the clinical SSFP cine images under failed $B_{0}$ shim can corrupt the image quality. Representative short-axis cine images with failed and successful $B_{0}$ shim are presented in (**A**). The banding artifact in the blood pool leads to severe imaging artifacts (arrows) and makes the images unreadable. (**B**) The signal intensity of the SSFP sequence in the presence of $B_{0}$ off-resonance (TR = 3.3 ms, phase cycle = 180 d).

**Figure 2 bioengineering-11-00210-f002:**
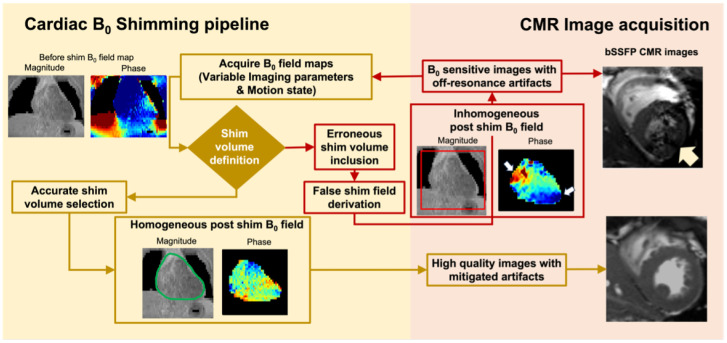
Cardiac $B_{0}$ shimming pipeline. A schematic flow chart of the CMR $B_{0}$ shimming workflow is presented. In today’s clinical practice, the manual selection of a shim box is used to derive the shim currents. The use of a rigid shim box can include undesired off-resonance fields outside of the heart and lead to failed $B_{0}$ shimming for $B_{0}$-sensitive CMR images.

**Figure 3 bioengineering-11-00210-f003:**
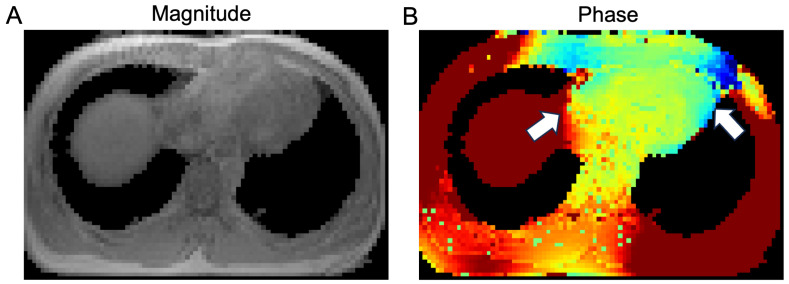
Magnitude and phase map from axial view. The magnitude map (**A**) image does not show a differentiable boundary as the white arrows show (the heart-liver boundary and the heart-lung interface), while the phase map (**B**) can provide the distinctive differences due to local frequency changes influenced by their respective location, shape and orientation in relation to both the air-filled space and the $B_{0}$ direction.

**Figure 4 bioengineering-11-00210-f004:**
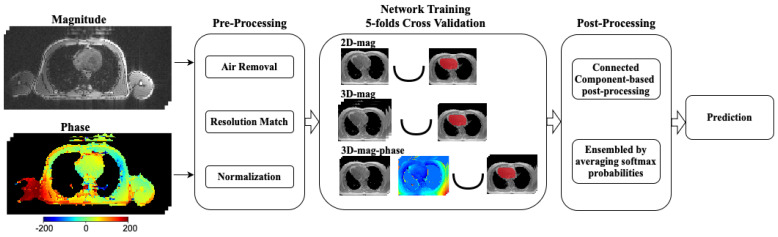
The general workflow of our experiments. The magnitude and phase maps are preprocessed and fed into several neural networks. We train 2D-mag-net, 3D-mag-net and 3D-mag-phase-net using 5-fold cross-validation. The segmentation outcomes are generated through the post-processing based on the output of the softmax layer.

**Figure 5 bioengineering-11-00210-f005:**
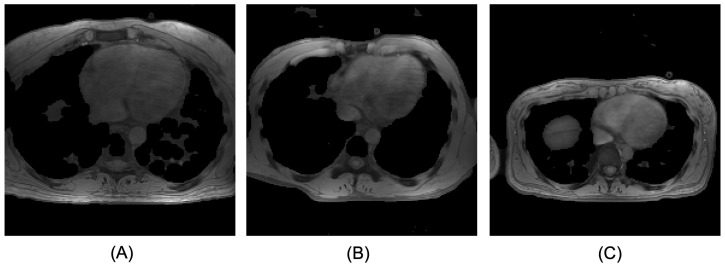
Axial views of normalized magnitude image. Subject (**A**) from back to chest distance is 229.69 mm, subject (**B**) is 203.13 mm and subject (**C**) is 154.12 mm. Various patient sizes exist and might affect the segmentation performance if it has not been addressed properly.

**Figure 6 bioengineering-11-00210-f006:**
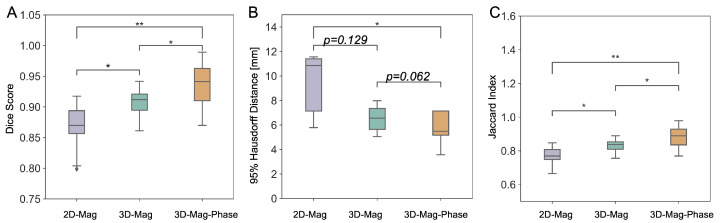
Comparative analysis of three different models on the T1-w dataset. The 3D-Mag-Phase net showed a significantly higher dice score than the other parameters (**A**) and a significantly higher Jaccard index than others (**C**). The 3D-Mag-Phase net is the only model that showed significant improvement In 95% HD compared to the conventional 2D model (**B**). (* indicates *p*-value < 0.05 and ** indicates *p*-value < 0.01.)

**Figure 7 bioengineering-11-00210-f007:**
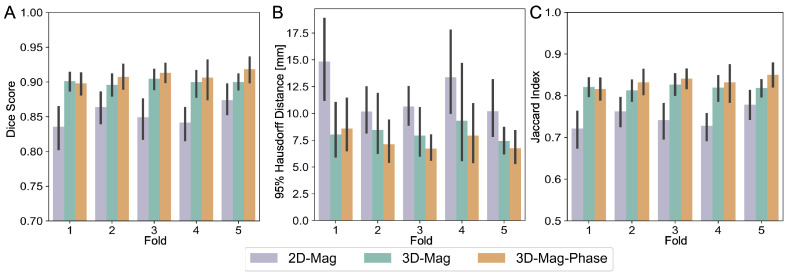
Comparison of different models on T1-w data among 5 folds in cross-validation. (**A**) shows the average dice score and standard deviation on T1-w test data, (**B**) shows the 95%HD, and (**C**) shows the Jaccard index.

**Figure 8 bioengineering-11-00210-f008:**
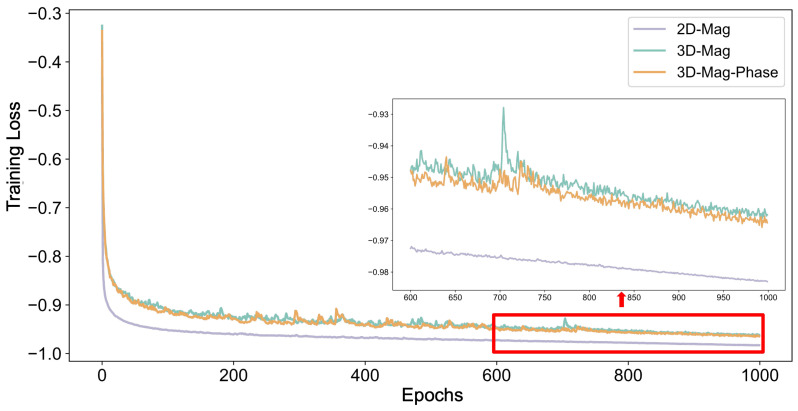
The training procedure of different models. This figure showed the training process of the 5th fold as an example.

**Figure 9 bioengineering-11-00210-f009:**
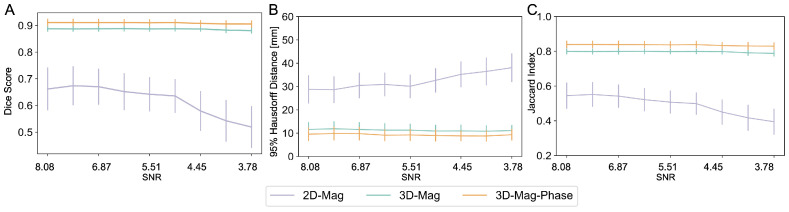
Model performance against field map SNR changes. In the 2D model, the segmentation performance was significantly reduced, corresponding to the increased noise level. On the contrary, the 3D models consistently performed with the SNR variation. In addition, the proposed dual-channel model (3D-mag-phase) demonstrates consistently improved segmentation results compared to the single-channel model (3D-mag). The (**A**–**C**) showed the corresponding segmentation performance in dice score, 95%HD and jaccard index.

**Figure 10 bioengineering-11-00210-f010:**
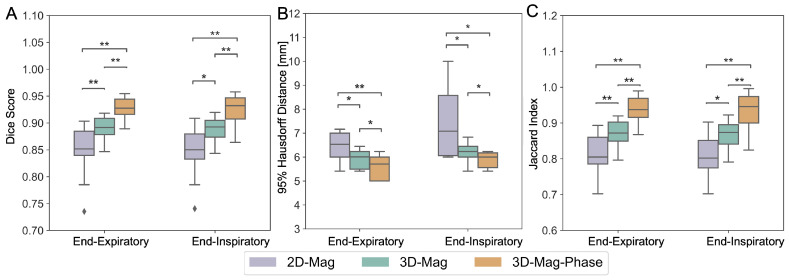
Comparison of different models on the proton-density weighted dataset. The dice score using different models in end-expiratory and end-inspiratory was shown in (**A**), 95%HD in (**B**) and jaccard index in (**C**).Our proposed 3D-mag-phase-net consistently and significantly outperformed others (* indicates *p*-value < 0.05 and ** indicates *p*-value < 0.01).

**Figure 11 bioengineering-11-00210-f011:**
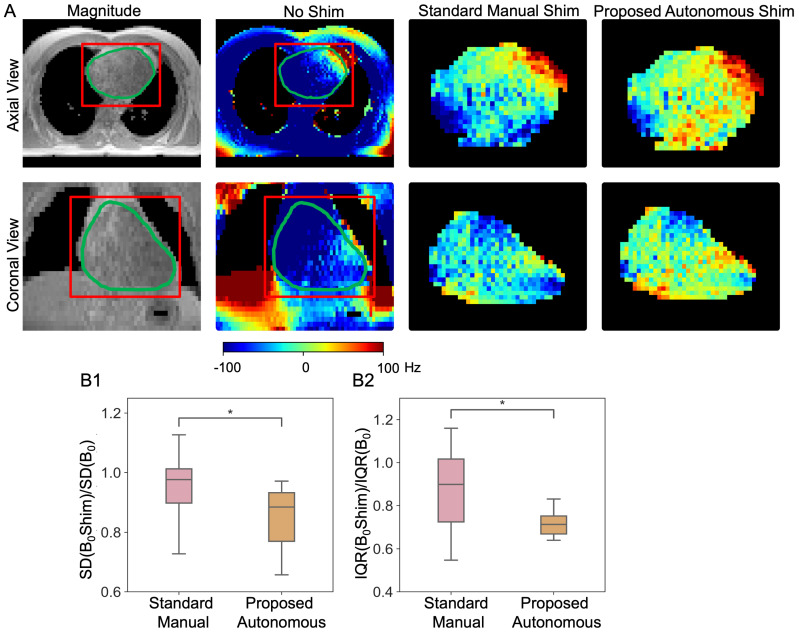
$B_{0}$ shimming comparisons between the manual box shim and autonomous contour shim. The representative figure (**A**) demonstrated the magnitude map, original phase map (No shim), and shimmed heart ROI using standard manual box shim and shimmed heart ROI using the proposed autonomous shim in axial and coronal view. The red square indicates the manual shimming box during standard manual shim, and the green contour indicates the auto-generated contour for our proposed method. The statistical results demonstrating the SD (**B1**) and IQR (**B2**) ratio after and before shimming, showed that our proposed method improved the shimming process significantly (* indicates *p*-value < 0.05).

**Figure 12 bioengineering-11-00210-f012:**
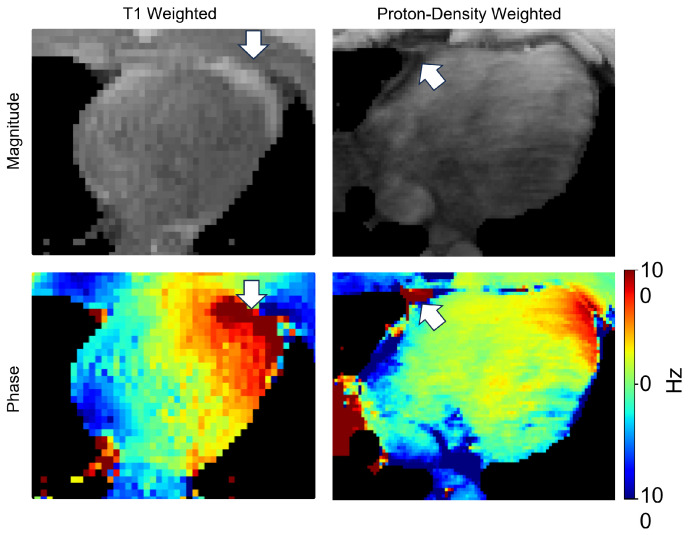
The contrast difference of fat region in different MRI parameter images. Magnitude images’ contrast changes on fat region when using different MRI parameters, as the white arrows show. However, the fat region is differentiable according to the phase maps.

**Table 1 bioengineering-11-00210-t001:** Data summary before and after proprocessing augmentation. The orientation, resolution and scaling augmentation performed in a preprocessing manner. Other augmentation methods not listed such as Gaussian noise, rotation, mirroring were performed on-the-fly with probability during training.

Aspects	Before Augmentation	After Augmentation
Orientation	RAS	RAS, LAS
Resolution	3.57 × 3.57 × 5.2 ${\mathrm{mm}}^{3}$	3.57 × 3.57 × 5.2 ${\mathrm{mm}}^{3}$, 4.46 × 4.46 × 5.2 ${\mathrm{mm}}^{3}$
Scaling	×1	×1, ×2, ×3

**Table 2 bioengineering-11-00210-t002:** The mean and standard deviation of segmentation performance on PDw data. The dice score, 95%HD and Jaccard index were reported under different respiratory cycle and for all the comparision, all *p* < 0.05.

Motion States	Model	Dice Score ↑	95%HD [mm] ↓	Jaccard Index ↓
End-Expiration	2D-Mag	0.85 ± 0.04	6.61 ±0.92	0.80 ± 0.06
	3D-Mag	0.89 ± 0.02	6.02 ±0.80	0.86 ± 0.03
	3D-Mag-Phase	**0.93± 0.02**	**5.78 ± 1.49**	**0.94 ± 0.03**
End-Inspiration	2D-Mag	0.83 ± 0.11	7.82 ± 2.42	0.78 ± 0.13
	3D-Mag	0.89 ± 0.02	6.40 ± 1.08	0.86 ± 0.04
	3D-Mag-Phase	**0.93 ± 0.03**	**5.92 ± 1.84**	**0.93 ± 0.05**

**Table 3 bioengineering-11-00210-t003:** The mean and standard deviation of dice scores using different types of models. The additional phase information can improve the model performance regardless of the model architecture, which highlights the importance of phase information. Additionally, our proposed model reported the highest dice score and reported a smaller SD (* indicates *p*-value < 0.05).

Models	Magnitude Only	Magnitude-Phase
SAM-Med3D	0.5814 (0.051)	-
U-Net	0.7988 (0.064)	0.8252 (0.049)
Swin UNETR	0.8571 (0.045)	0.8623 (0.044)
Ours (3D-mag-phase-net) *	0.9065 (0.023)	**0.9379 (0.038)**

## Data Availability

Data available on request due to restrictions of privacy. The data presented in this study are available on request from the corresponding author.

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
