# Peer review of "Reliable Off-Resonance Correction in High-Field Cardiac MRI Using Autonomous Cardiac B_0_ Segmentation with Dual-Modality Deep Neural Networks"

_bioengineering, 2024, doi:10.3390/bioengineering11030210_

Round 1
Reviewer 1 Report
Comments and Suggestions for Authors
Authors made a good attempt to enhance the cardiac segmentation for B0 shimming.
1. The paper is well written and abstract is good.
2. The dataset details before augmentation and after augmentation can be included as a table for better understanding.
3. Comparison can be made with the state of art literature to claim the superiority.
4. The IoU index also can be experimented to claim the superiority of the segmentation.
5. Conclusion can be included as a separate section and limitations can be addressed.
Comments on the Quality of English LanguageMinor editing is required
Reviewer 2 Report
Comments and Suggestions for Authors
This paper proposes a dual-modality DNN for cardiac B0 segmentation, which aims for reliable off-resonance correction in high-field cardiac MRI. The experimental results show the effectiveness of the proposed method. Some comments can be found as follows:
1. The dual modality only appears in the title, which seems to be the main characteristic of the proposed network. Please clarify this.
2. The main novelty and contributions of this work is unclear, better to summarize these point by point at the end of the introduction part.
3. A related work section which is different from the introduction would be helpful for readers to understand the background development.
4. The datasets are suggested to be stated in the experimental section instead of the methodology. Please improve the organization and presentation quality.
5. It is suggested to add some descriptions of the transformer compared to traditional DNN and the reasons of not adopting the transformer. Some literature include RTN: Reinforced transformer network for coronary CT angiography vessel-level image quality assessment, Attention transformer mechanism and fusion-based deep learning architecture for MRI brain tumor classification system, etc.
6. In the experiments, an ablation study to verify each key component of the proposed DNN shall be added. And there lack the comparisons between the proposed method with state-of-the-arts.
Comments on the Quality of English Language
N/A
Reviewer 3 Report
Comments and Suggestions for Authors
Magnificent article. Very well explained and contextualized. With very good results compared with different own and state of the art models.
To put some but, I think the following issues should be addressed:
In section 2.2 on image preprocessing, it is stated that the background has been removed, but without explaining too much how exactly. From the reference to Otsu it is understood that a thresholding is done. I think it would be interesting if it were explained a little better how exactly this preprocessing is done and even showing some graphical example of the improvement introduced. Similarly, in the case of noise standardization, I think it would be interesting to explain the process a little better and show some example. Is it done with the mean and standard deviation per case or of all the cases under study?
Part of the discussion and future work I think should be put in the conclusions of the paper, so that these are somewhat more extensive and clear.
Minor issues:
If possible, figure 5 should be shown before section 2.6 and not in the middle of this one....
The results shown in section 3.1, I think, would look better if they were shown in tabular form.
If possible, Figure 8 should be shown before section 3.2.
The results in section 3.2.2, I think, would be better displayed in tabular form.
In the tables, the best result should be bolded. Also in table 1.
The results shown in section 3.2.4, I think, would be better displayed in a table.
Reviewer 4 Report
Comments and Suggestions for Authors
This is an interesting, well written paper. The methodology sounds. Authors proposed deep segmentation architecture based on popular U-net encoder-decoder. Authors developed a dual-channel model combined with U-Net architecture to accommodate the potential variation of field map acquisition. Authors tested the model at 3.0T, where the B0 off-resonance is stronger and affects its daily application in the clinical setting.
Detailed comments:
1) Please write directly in the introduction what is the innovation introduced in your work in relation to the methods published so far
2) “As we used the cross-fold validation in training (…)” please give more details about training and add them to section 2.4
3) “denote the sets of heart voxels in ground-truth” – how did you obtain ground-truth data?
4) Please give more details about implementation of the proposed NN.
5) Please publish source codes in open repository for example github with pertained weights (data is already available on request as you stated). Without source codes + data experiments are virtually impossible to reproduce results.
6) In line 184, 217 or 220 there is a comma at the beginning of the line, in line 182 there is additional dot
Round 2
Reviewer 1 Report
Comments and Suggestions for Authors
Authors addressed all my previous concerns well.
Reviewer 2 Report
Comments and Suggestions for Authors
The authors have addressed my comments.
Comments on the Quality of English LanguageN/A
Reviewer 4 Report
Comments and Suggestions for Authors
The authors addressed my remarks. In my opinion, the paper can be accepted as it is.